# ZIF-8/Chitosan Composite Hydrogel as a High-Performance Separator for Bioelectrochemical Systems

**DOI:** 10.3390/membranes15090282

**Published:** 2025-09-22

**Authors:** Henry Pupiales, Raúl Bahamonde Soria, Daniel Arboleda, Carlos Cevallos, Christian Alcívar, Laurent Francis, Xiao Xu, Patricia Luis

**Affiliations:** 1Renewable Energy Laboratory, Faculty of Chemical Sciences, Central University of Ecuador, Quito 170521, Ecuador; hdpupiales@uce.edu.ec (H.P.); dgarboleda@uce.edu.ec (D.A.); 2Faculty of Chemical Sciences, Central University of Ecuador, Quito 170521, Ecuador; cacevallosm@uce.edu.ec (C.C.); cdalcivar@uce.edu.ec (C.A.); 3Institute of Information and Communication Technologies, Electronics and Applied Mathematics, UCLouvain, Place Sainte Barbe 2, 1348 Louvain-la-Neuve, Belgium; laurent.francis@uclouvain.be; 4Institute of Mechanics, Materials and Civil Engineering-Materials & Process Engineering (iMMC-IMAP), UCLouvain, Place Sainte Barbe 2, 1348 Louvain-la-Neuve, Belgium; xiao.xu@gold.kobe-u.ac.jp (X.X.); patricia.luis@uclouvain.be (P.L.)

**Keywords:** chitosan, nano enhancer, chitosan/ZIF-8 composite, membrane separator, bioelectrochemical systems

## Abstract

Chitosan biopolymer membranes reinforced with channel-selective ZIF-8 nanofillers were developed and thoroughly characterized as separators for bioelectrochemical systems. This study explores the synergistic effect of incorporating ZIF-8 into a chitosan matrix to enhance membrane performance. Key properties including water retention, chemical and thermal stability, surface resistance, antifouling capacity, and ionic conductivity were evaluated and benchmarked against commercial Nafion-117 and nanofiltration (NF) membranes. The ZIF-8/chitosan composite membranes (ZIF-8/CS) demonstrated excellent water retention and structural stability under harsh conditions, along with significantly reduced surface resistance and effective rejection of organic contaminants and salts (NaCl, Na_2_SO_4_). Notably, the composite ZIF-8/CS membranes achieved an ionic conductivity of 0.099 S/cm, approaching the value of Nafion-117 (0.13 S/cm) and substantially surpassing that of the NF membrane (0.013 S/cm). These results indicate that ZIF-8-reinforced chitosan membranes present a promising, sustainable, and cost-effective alternative to traditional separators in bioelectrochemical applications.

## 1. Introduction

The generation of wastewater from domestic, agricultural and industrial sources has a major impact on aquatic ecology, drinking water quality and public health [1,2]. Current wastewater treatment processes necessitate large capital investments and these systems emit substantial amounts of greenhouse gases into the atmosphere [3,4,5]. The bio-electrochemical systems are green technology to address this problem, because these devises transform organic compounds into fuels, electric energy or hydrogen [6,7]. Two-chamber microbial fuel cells (MFCs) utilize a proton exchange membrane (PEM) or a separator to divide the anode and cathode compartments. Ion exchange membranes have positive or negative functional groups that can facilitate selective transport of opposing ions. In bioelectrochemical systems (BES), the PEM acts as a physical barrier that separates the anodic and cathodic reactions catalyzed by microorganisms while maintaining chemical gradients and reducing the distance between electrodes [8]. An ideal separator is engineered to enhance energy production, minimize oxygen crossover to the anode, restrict the diffusion of fuel and organic matter to the cathode, create no resistance to the transport of ionic species, prevent biofouling of the cathode, and expand the functionality of the BES beyond power generation [9,10,11,12].

Nafion-117 is a membrane widely used in BES such as microbial fuel cells (MFC) and microbial electrolysis cells (MEC). Nafion-117 membranes offer excellent mechanical stability, high cationic conductivity and resistance to chemicals [13]. However, its perfluorinated sulfonic acid groups tend to deprotonate in alkaline environments. Under high pH conditions (typically above pH 10), the Nafion membrane may undergo structural destabilization and chemical degradation, which reduces its ionic transport capacity. In addition, Nafion-117 membranes account for approximately 40% of the total cost of BES, with an estimated market price of around 2000 USD/m^2^, making them economically limiting for large-scale applications. Their disposal also presents significant environmental concerns [14]. Similarly, polyamide nanofiltration (NF) membranes, which have not previously been tested in microbial fuel cells (MFCs), pose similar challenges. NF membranes are chemically stable within a pH range of approximately 3 to 10, and are able to retain small organic particles and salts, potentially reducing oxygen crossover. Although they may introduce higher ion transfer resistance than the ion-selective membranes commonly used in MFCs, their lower cost estimated at 20 to 25 USD/m^2^ makes them promising candidates for evaluation as separators in these systems due to their unique properties and potential advantages [12,15].

Recent research has focused on integrating hybrid organic-inorganic membranes into a single membrane to take advantage of the complementary properties of each component [16]. Natural polymers such as chitosan are a sustainable and cost-effective alternative to synthetic polymers. Chitosan-based hybrid composites, nanomaterials, ion exchange membranes, and mixed matrix membranes have attracted extensive attention [17,18,19]. Zeolitic imidazolate frameworks (ZIFs), a subclass of metal–organic frameworks (MOFs), are promising materials due to their high surface area, crystallinity, and microporous structure. In particular, ZIF-8, which has large pores (11.6 Å) that are accessible through small openings (3.4 Å), suggests potential applications in various fields [20,21].

This study presents a novel separator based on a chitosan biopolymer combined with nanochannels formed by ZIF-8 nanofillers. The composite membranes were thoroughly characterized and benchmarked against commercial membranes (Nafion-117 and NF) to evaluate their potential for use in bioelectrochemical systems (BES). The incorporation of ZIF-8 nanofillers effectively restructures the chitosan matrix, facilitating selective ion transport through newly formed nanochannels. This enhancement results in a high proton conductivity of up to 0.09 S/cm at room temperature and low surface resistance. Additionally, fouling behavior was assessed using bovine serum albumin (BSA) to simulate protein-related membrane fouling. These promising results demonstrate the potential of ZIF-8/chitosan composite membranes as cost-effective, efficient separators for BES applications, motivating further investigation into their practical implementation.

## 2. Materials and Methods

### 2.1. Materials

Sulfuric acid (H_2_SO_4_, 98%), sodium hydroxide (NaOH, 97%), sodium chloride (NaCl, 99.99%), anhydrous sodium sulfate (Na_2_SO_4_, ≥99.0%), hydrochloric acid fuming (HCl, 37%), potassium chloride (KCl, ≥99.5%), iron (II) sulfate heptahydrate (FeSO_4_·7H_2_O, 99.5%), hydrogen peroxide (H_2_O_2_, 30%), magnesium sulfate heptahydrate (MgSO_4_.7 H_2_O, 99.0%) and acetic acid (CH_3_COOH, 99.8%), were purchased from Merck KGaA, Darmstadt, Germany. Zinc nitrate hexahydrate (Zn (NO_3_)_2_⋅6H_2_O, 99%), 2-Methylimidazole (Hmim, 97%) and Dimethyl carbonate (DMC, 99%) were purchased from Alfa Aesar, Karlsruhe, Germany. The Bovine Serum Albumin (Lyophilized Powder BSA ~66 kDa) and Chitosan CS polymer powder (low molecular weight) were obtained from Sigma-Aldrich, St. Louis, MO, USA. Triethylamine (TEA, N(CH_2_CH_3_)_3_, 99.5%), ethanolamine (NH_2_CH_2_CH_2_OH, 99.5%), absolute ethanol (C_2_H_5_OH, 99.9%), Nafion-117 and NF membrane were provided by VWR International, Leuven, Belgium. Polyacrylonitrile-PX-FSR-1 (PAN) was purchased from Synder Filtration, Vacaville, CA, USA. All chemicals were used as received without further purification procedure.

### 2.2. Synthesis of ZIF-8 Particles

ZIF-8 particles were synthesized according to the method described in [22]. Initially, 2.00 g of Zn (NO_3_)_2_⋅6H_2_O was dissolved in 12.1 g of deionized water. Furthermore, 3.31 g of Hmim was dissolved in 48.4 g of deionized water. Thereafter, 3.00 mL of TEA was added to the Hmim solution. Then, the two solutions were mixed for 30 min. The product was then centrifuged at 4000 rpm for 15 min and washed twice with deionized water and once with absolute ethanol. Finally, the product was dried in an oven at 120 °C for 10 h before the following use.

### 2.3. Membrane Preparation

The composite ZIF-8/CS membrane were synthesized on a PAN support according to the method described in [22]. First, 1.00 g of CS was dissolved in 49.0 g of 2.00 %wt. acetic acid solution and stirred at room temperature for 2 h. Then, the pH of the solution was adjusted to ≥4 using ethanolamine [23]. The solution was stirred for 24 h. Thereafter, 0.150 g of ZIF-8 nanoparticles, a concentration effective in previous pervaporation studies, was added to optimize porosity and ion transport in the membrane for BES. The mixture was stirred for 12 h to ensure uniform distribution of the particles in the CS. After stirring, the membranes were prepared by solvent evaporation method. Briefly, the solution was poured onto support membranes (PAN) with a stainless-steel retaining ring and dried in a fume hood to obtain the dried membrane. Prior to casting, the PAN support was hydrolyzed in 2.00 wt% NaOH at 60 °C for 30 min and rinsed with deionized water to neutral pH. After drying, composite ZIF-8/CS membrane was neutralized with NaOH aqueous solution. For the method, NaOH was dissolved in deionized water to make 2.00 %wt NaOH aqueous solution. Composite ZIF-8/CS membrane was immersed in this solution for three times (15 min each time) and then washed with absolute ethanol for three times [24]. The active layer thickness of the membranes was measured using a digital micrometer (Mitutoyo Corporation, Kawasaki, Japan), yielding values between 36 and 43 µm. The total membrane thickness, including the PAN support, was approximately 120 µm. Finally, the membranes were stored in a 0.500 g/L NaCl solution before the next use.

### 2.4. Membrane Characterization

A scanning electron microscope (SEM) (Carl Zeiss Microscopy GmbH, Oberkochen, Germany) was used to analyze the ZIF-8 nanofillers and show the morphology of the composite ZIF-8/CS membrane. The membranes were previously immersed in liquid nitrogen to obtain a clean cross section. Then, a thin layer of gold was deposited under vacuum with a sputter coater (BALZERS SCD 030; Balzers Union, Balzers, Liechtenstein) to make the samples conductive. As for the crystallinity of ZIF-8 nanofillers, pure CS and the composite ZIF-8/CS membrane, it was investigated by X-ray diffraction (Bruker D8 Advance diffractometer (Bruker AXS GmbH, Karlsruhe, Germany) with Cu Kα radiation (λ = 1.54 Å)) in the range of 6.00 to 60.0°, while the FT-IR spectra of the ZIF-8 nanofillers and composite ZIF-8/CS membranes. were obtained using a Bruker Optik GmbH, Ettlingen, Germany from 400 cm^−1^ to 4000 cm^−1^ in ATR mode to investigate the chemical composition of the fabricated membrane.

### 2.5. Water Uptake Test

Water absorption tests on the membranes were carried out gravimetrically using an analytical balance (Denver Instrument Company, Arvada, CO, USA). Before conducting water uptake, membranes were dried into an oven at 80 °C for 12 h to remove excess humidity. Afterward, the samples were cut into 2 × 2 cm squares. After cutting, the dried membrane samples were immersed in deionized water for 24 h at 37 °C. Hydrated membrane samples (W_wet_) were weighed as quickly as possible. After weighing, the membrane samples were dried again at 60 °C for 24 h in an oven to evaporate all of the adsorbed water. The membrane samples were reweighed (W_dry_). All experiments were performed in triplicate. Water absorption was calculated using the following equation [25,26]:(1)$$Water\;uptake,\;\%=\frac{M_{wet}-M_{dry}}{M_{dry}}\times 100\%$$

In addition, contact angle measurements were performed to evaluate the surface wettability of the composite membranes. A contact angle goniometer was used, and distilled water droplets of 5 μL were carefully placed on the membrane surface.

### 2.6. Durability Test

The durability of the membranes was evaluated by the Fenton test [27]. First, the samples were cut into 2 × 2 cm squares. After cutting, the dried membrane samples were immersed into 50.0 mL Fenton solution (3.00 %wt. hydrogen peroxide solution and 20.0 ppm Fe^2+^) for 150 h at 80 °C [28]. Finally, the weights of dried samples before (Wa) and after (Wd) the experiment were compared. Membrane durability was calculated from the following equation [29,30]:(2)$$Membrane\;durability\;,\;\%=\frac{\left(W_{d}-w_{a}\right)}{W_{a}}\times 100\%$$

### 2.7. Surface Electrical Resistance

The surface electrical resistance of the membrane was measured by chronopotentiometry. The diagram of the experimental cell is shown below (Figure 1a).

The four-chamber test cell (Figure 1a) used had an effective membrane area (S) of 7.00 cm^2^ and two Ag/AgCl reference electrodes, closed to the membrane, to measure the potential difference across the membrane. After the cell was assembled, solutions of NaCl 0.100 molL^−1^, Na_2_SO_4_ 0.200 molL^−1^ and bovine serum albumin (BSA) 0.500 gL^−1^ were circulated in the corresponding chambers at a specific flow rate using four pumps. Then, a direct current (I) of 11.0 mA and potential of 9.00 V was applied for 24 h. At the same time, the potential difference without the membrane sample (U = 0.03 V) and with the membrane sample (U_o_) was measured using a voltage data logger (Pico Technology, St Neots, Cambridgeshire, United Kingdom) [31,32]. Electrical resistance was calculated from the following equation.(3)$${Electrical\;resistance,\;R}_{mem}=\left(\frac{U-U_{0}}{I}\right)\times S$$

### 2.8. Anti-Fouling Analysis

The antifouling performance of the membranes was evaluated by observing the changes in surface morphology using an atomic force microscope (Park Systems Corporation, Suwon, South Korea). The samples were initially cut into approximately 1.00 cm squares. The scan size for AFM imaging was 2.5 × 2.5 μm. The average roughness (Ra) of the membranes before and after biofouling was then calculated to evaluate the changes in surface texture [33,34].

### 2.9. Impedance Spectroscopy Characterizations

Impedance spectroscopy (EIS) measurements were performed using a custom-made Teflon four-electrode cell equipped with platinum electrodes, as illustrated in Figure 1b. Each membrane was pre-hydrated in deionized water for 24 h and placed horizontally inside the cell, maintaining full hydration during the measurements. The distance between adjacent electrodes was 10 mm. All membranes were immersed in 0.01 M HCl bath solution before the measurements. A Metrohm Autolab PGSTAT302N potentiostat/galvanostat with a frequency response analyzer was used to apply a 10-mV sinusoidal AC voltage over a frequency range from 0.1 Hz to 100 kHz. Nyquist plots were fitted using the equivalent circuit shown in Figure 1b, typically represented as R(QR), using ZSimpWin software version 3.20 (Princeton Applied Research, Oak Ridge, TN, USA). In this model, Rs represents the intrinsic series resistance of the system. Rm corresponds to the membrane resistance, and Q is a constant phase element (CPE) used to account for non-ideal capacitive behavior in the interface, which may arise from surface roughness, heterogeneity, or distributed charge relaxation times. The ionic conductivity in the in-plane direction was calculated using the equation:(4)$$Ionic\;Conductivity,\;\sigma =\frac{l}{Rm.d.w}$$
where l is the distance between the current-carrying electrodes, Rm is the membrane resistance, d is the membrane thickness, and w is the membrane width.

In addition to the measurements at room temperature, the ionic conductivity of the membranes was evaluated at different temperatures (40 °C, 60 °C, and 80 °C). These temperature-dependent measurements were used to construct Arrhenius plots, which allowed the calculation of the activation energy (Ea) using the following equation:(5)$$\sigma =\frac{\sigma_{0}}{T}exp(-\frac{Ea}{RT})$$
where σ is the conductivity, σ_o_ is a pre-exponential factor, Ea—activation energy, R—the gas constant and T—the absolute temperature.

## 3. Results and Discussion

### 3.1. Membrane Characterization

#### 3.1.1. Characterization of ZIF-8

Figure 2a shows the SEM images of the synthesized ZIF-8 particles.

The size of the synthesized ZIF-8 nanoparticles was in the order of 100–200 nm. In addition, small size crystals as nanofillers within the chitosan polymer matrix can reduce the non-selective volumes at the nanofiller/polymer interface, thus high membrane selectivity can be achieved [35]. ZIF-8 is also known for its high surface area, with reported values ranging from 428 to 490 m^2^/g, which contributes significantly to its performance in separation and ion transport applications [36]. XRD analysis showed a crystal structure of ZIF-8 with characteristic peaks at 2θ values of 7.3°, 10.3°, 12.7°, 14.7°, 16.4°, 18.0°, 22.1°, 24.4°, 25.5°, 26.6° 29.6° and 34.9°, as shown in Figure 2b. The XRD patterns of the synthesized ZIF-8 are in good agreement with those in the literature [36,37,38], indicating that a very crystalline ZIF-8 without impurities was obtained.

#### 3.1.2. Morphology of the Membranes

The ZIF-8/CS composite membranes were characterized by SEM. As shown in Figure 2c, the image corresponds to the flat surface of the membrane. It displays a smooth surface with some defects. Moreover, the ZIF-8 particles appear to be well dispersed within the CS matrix, with no apparent cracks or phase separation, indicating good interfacial adhesion. This suggests good compatibility and a uniform interfacial distribution between the ZIF-8 nanofillers and the CS matrix.

#### 3.1.3. XRD Characterizations of ZIF-8/CS Composite Membranes

An XRD spectrum for the composite ZIF-8/CS membrane (Figure 2b) shows the characteristic diffraction peak of CS located at 2θ = 21.9° [39], demonstrating that the incorporation of ZIF-8 nanofillers did not influence the CS crystal form. Moreover, the composite ZIF-8/CS membrane shows a new peak located around 2θ = 32.1°, indicating the presence of ZIF-8 nanofillers within the overall polymeric matrix [40].

#### 3.1.4. FTIR Spectroscopy Analysis of Membranes

The chemical functional groups of ZIF-8 nanofiller, pure CS and composite ZIF-8/CS membrane were evaluated by FTR (Figure 2d). Significant bands were observed at 420, 692, 760, 993, 1146, 1178, 1310, 1357, 1456, 1587, 2799, 2929 and 3134 cm^−1^ for the ZIF-8 particulate samples. The peak at 2929 cm^−1^ was attributed to asymmetric C-H aliphatic stretching vibration, while the peak at 1587 cm^−1^ is due to -C=N stretching vibration. The peak at 760 cm^−1^ was associated with the bending vibration of the imidazole ring. The Zn-N stretching vibration was observed at 420 cm^−1^, implying that the zinc ions chemically combined with the nitrogen atoms of the methylimidazole groups to form the imidazole ring [41]. The characteristic peaks in the spectrum of the ZIF-8 nanofiller appeared in the FTIR spectrum of the composite ZIF-8/CS, demonstrating that the ZIF-8 nanofiller was successfully incorporated into the CS matrix.

### 3.2. Water Uptake of the Membrane

The values of percent water absorption of the membranes for 24 h were 161 ± 1.2%, 144 ± 0.37% and 21.8 ± 0.2% for the samples of composite ZIF-8/CS membrane, NF and Nafion-117, respectively (Figure 3a).

The composite ZIF-8/CS membrane showed a high level of hydration that can be attributed to the combination of the porous structure of ZIF-8 and the hydrophilic properties of chitosan. The -OH and -NH_2_ functional groups present in chitosan can form hydrogen bonds with water molecules, thus increasing the water holding capacity [42,43].

This behavior was supported by the contact angle measurement, where the composite membrane presented a value of 63°, indicative of a more hydrophilic surface compared to the commercial nanofiltration membrane (68°) and Nafion-117 (88°). The NF membrane showed considerable water absorption capacity, making it particularly suitable for applications in BES. In contrast, the Nafion-117 membrane showed low water sorption capacity. Despite its high ionic conductivity, the lack of a suitable porous structure may restrict its effectiveness in processes that depend on high hydration [44]. This difference in water absorption capacity highlights the importance of considering membrane microstructure when selecting materials to optimize membrane performance.

### 3.3. Durability Test Results

Nafion-117, NF and composite ZIF-8/CS membranes showed 100 ± 0.2%, 99.4 ± 0.5% and 96.0 ± 1.0% of oxidation resistance, respectively, after exposure to Fenton’s reagent (Figure 3b).

Although both composite ZIF-8/CS and NF membranes exhibited comparable stability, their resistance to oxidation was inferior compared to Nafion-117. Nevertheless, the membrane matrix remained intact after 192 h under extreme conditions. This stability is probably related to the structural robustness of ZIF-8 and the chemical characteristics of chitosan as previously reported in the literature [29,45]. These properties help protect the membrane from peroxyl radicals generated during the operation of the bioelectrochemical system (BES), potentially extending its service life [46,47].

### 3.4. Surface Electrical Resistance and Anti-Fouling Evaluation

We found that surface electrical resistance values for the active area of the membrane of: 4.91 Ωcm^2^ for composite ZIF-8/CS membrane, 5.03 Ωcm^2^ for Nafion-117 and 37.1 Ωcm^2^ for NF. The composite ZIF-8/CS membrane has the lowest surface electrical resistance among three kinds of membranes. This difference could be attributed to the materials present in the mixed matrix membrane structure: the composite ZIF-8/CS membrane contains ZIF-8 nanofillers, a very porous material that could increase the active surface area of the membrane by better distributing the ions in the CS matrix. Moreover, they can act as an efficient channel for ion transport, improving the performance of the composite ZIF-8/CS. In addition, CS exhibits good ionic conductivity in wet media due to the presence of amino (-NH_2_) and hydroxyl (-OH) groups in its structure, which can protonate in acidic conditions, facilitating the transport of protons (H^+^) in contact with the membrane matrix [12,48,49].

The antifouling property of the membranes was evaluated by the variation in the electrical resistance of the membranes as a function of time, as shown in Figure 4.

Fouling is caused by the accumulation of organic molecules and salts on the surface of the membrane or within its pores [50]. In our experiments, bovine serum albumin (BSA) was used as an organic contaminant. Organic BSA has a volume of 84.5 nm^3^ and a slight negative density of charge, which increases its mobility in solution and its probability of collision on the membrane surface [51]. The results suggest that there is a slight increase in the surface electrical resistance of the membranes when they come into contact with BSA (Figure 4b). Conversely, no significant variation in electrical resistance was observed over time when exposed to salts (Figure 4a), suggesting minimal fouling by salts. This increase in electrical resistance is likely attributed to membrane surface fouling. In the case of membranes composed with chitosan, this polymer acts as an antifouling barrier due to its ability to form a surface hydration layer that repels proteins, in addition to generating electrostatic repulsion with BSA molecules due to their positive charge under slightly acidic or neutral conditions. These effects reduce protein adsorption and thus fouling. This behavior has been supported in previous studies showing that modification of membranes with chitosan improves flux recovery and reduces protein adsorption, as observed by lower variations in roughness and maximum height after fouling (Figure 5 and Figure 6) [52].

In addition, 3D images obtained by AFM were used for a detailed investigation of the surface morphology before and after fouling of the membranes with BSA (Figure 5 and Figure 6). The data obtained reveal remarkable differences in the behavior of the ZIF-8/CS membrane compared to the other membranes analyzed.

Table 1 presents the mean roughness (Ra) and maximum height (Sz) values for the three membranes analyzed previously. A significant increase in Sa and Sz was observed for all membranes after fouling, as shown in Figure 5 and Figure 6. In particular, the composite ZIF-8/CS membrane, although showing a minor absolute change compared to the Nafion-117 and NF membrane, shows an increase of 25.88% in Sa and 26.02% in Sz. Although these changes are less compared to those observed in Nafion-117 (179.48% in roughness) and NF (117.12%), they indicate that the ZIF-8/CS composite membrane is more resistant to fouling by BSA. The lower susceptibility of the ZIF-8/CS composite membrane to fouling suggests that its surface properties are more effective in reducing protein adsorption. This not only improves its performance, but also implies greater viability for long-term applications. These height variations provide insight into the surface roughness and morphology of the membranes. Before fouling, the membrane surfaces exhibit a more homogeneous distribution of elevations, with well-defined peaks and valleys, which are typically associated with enhanced surface area and potential for improved mass transfer. However, after fouling, an increase in the density of yellow (elevated) regions is observed, indicating the accumulation of foulants on the surface. At the same time, the reduction or masking of black regions suggests that valleys become filled, resulting in a smoother but more obstructed surface. This change in topography is indicative of pore blockage and surface coverage by foulants, which can negatively affect the membrane’s performance by reducing permeability and increasing resistance to transport [51,53,54].

### 3.5. Impedance Spectroscopy

In the Nyquist plots (Figure 7), a single semicircle is observed for all membranes, which indicates that the ionic conduction can be effectively modeled using a simplified equivalent circuit of the form R(QR).

At high frequencies, the impedance spectra show a characteristic resistance–capacitor loop corresponding to the bulk response of the membrane material. Quantitatively, the Nafion-117 membrane exhibited an ionic conductivity of 0.13 S/cm, while the ZIF-8/CS composite membrane showed a comparable conductivity of 0.09 S/cm. In contrast, the nanofiltration membrane presented a significantly lower conductivity of 1.3 × 10^−2^ S/cm. The proximity of the ZIF-8/CS membrane’s conductivity to that of Nafion-117 highlights its potential as an efficient ion conductor (Table 2) [55]. Furthermore, the increased ionic conductivity observed in the composite ZIF-8/chitosan membrane can be attributed to several synergistic mechanisms arising from the interaction between the ZIF-8 nanofillers and the chitosan matrix. First, the incorporation of ZIF-8, a highly porous material with well-defined micropores and a high specific surface area, provides additional selective ion transport pathways within the membrane structure, thereby enhancing ionic activity and facilitating ion migration. Second, the inherently hydrophilic nature of both ZIF-8 and the chitosan matrix—with its abundance of amino (-NH_2_) and hydroxyl (-OH) groups—improves water uptake and retention within the membrane. This hydration is crucial for maintaining a conductive medium, as it supports proton hopping and the mobility of hydrated cations. Finally, at the interface between ZIF-8 particles and chitosan, favorable interactions likely promote the formation of active sites that facilitate ion hopping mechanisms, further contributing to the enhanced ionic conductivity. Together, these synergistic effects result in a composite membrane with significantly improved ion transport properties compared to pure chitosan membranes [22,56].

To further support these observations, the temperature-dependent ionic conductivity was analyzed and plotted in Figure 8a, while the Arrhenius behavior ln(σ.T) vs. 1000/T) is shown in Figure 8b. From the linear fitting of the Arrhenius plots, activation energy (Ea) values were extracted for each membrane. The corresponding Arrhenius plot (Figure 8b) revealed Ea values of ∼4.50 and ∼6.22 kJ/mol for the Nafion-117 and composite ZIF-8/CS membranes, respectively. Although the composite ZIF-8/CS membrane shows a slightly higher Ea, this is consistent with a transport mechanism based on interfacial ion hopping, which requires an initial activation step. Once this barrier is overcome, the membrane provides efficient ion conduction pathways, as evidenced by its high conductivity and the positive temperature response shown in Figure 8a, where a clear increase in the ionic conductivity of the composite ZIF-8/CS membrane with increasing temperature supports its efficient ion transport behavior [57].

A broader comparison of key properties—including water uptake, surface resistance, ionic conductivity, and antifouling performance—is presented in Table 3. This comparison, based on data from the literature on both commercial and lab-developed membranes, highlights the superior performance of the ZIF-8/CS membrane. Its high-water uptake and low surface resistance translate into enhanced ionic conductivity, while its strong antifouling behavior and oxidative stability suggest excellent long-term performance in bioelectrochemical systems.

## 4. Conclusions

The ZIF-8/chitosan (CS) composite membrane exhibited an ionic conductivity comparable to that of Nafion-117 (0.099 S/cm vs. 0.13 S/cm), indicating good ion transport performance. In addition to its conductivity, the ZIF-8/CS membrane demonstrated high water retention, excellent thermal and oxidative stability, low surface resistance, and intrinsic antifouling properties. Its biodegradability, lower cost, and sustainable origin make it a promising alternative for bioelectrochemical systems (BES), particularly in applications where environmental sustainability and material affordability are important. Although studies on its direct application in BES are still limited, these results support its potential use in microbial fuel cells, desalination processes, CO_2_ capture, and water purification technologies.

Future work will focus on assessing its performance in real MFC systems to further validate its practical applicability under operational conditions. Future work will include testing its energy generation performance in actual MFC systems to further validate its practical potential.

## Figures and Tables

**Figure 1 membranes-15-00282-f001:**
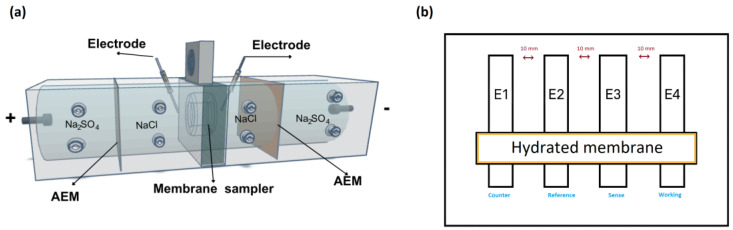
(**a**) Schematic representation of the experimental setup. Anion-exchange membrane (AEM) graphite–ruthenium electrodes (+, −); (**b**) schematic diagram of the four-electrode cell used for electrochemical impedance spectroscopy (EIS) measurements.

**Figure 2 membranes-15-00282-f002:**
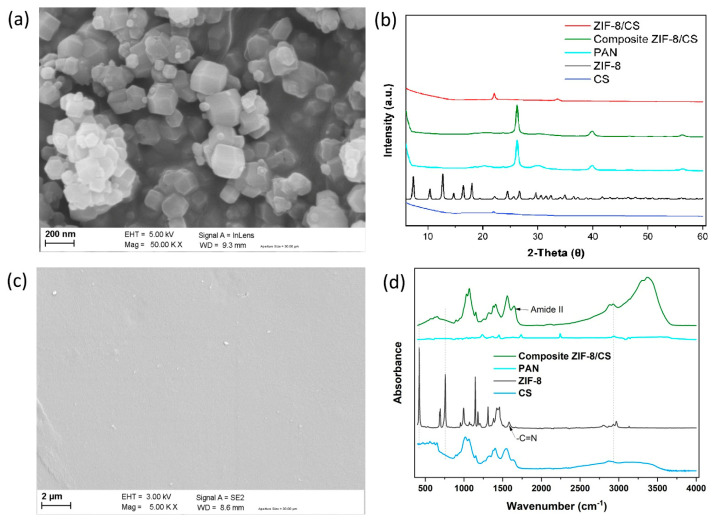
(**a**) SEM image of the synthesized ZIF–8 nanoparticles. (**b**) XRD patterns of composite ZIF–8/CS with their different components. (**c**) SEM image of the Composite ZIF–8/CS membrane. (**d**) FTIR spectra of composite ZIF–8/CS membrane with their different components.

**Figure 3 membranes-15-00282-f003:**
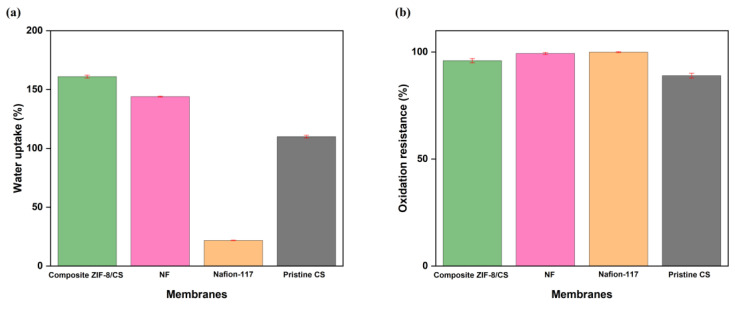
(**a**) Percent water absorption of membrane samples for 24 h 37 °C. (**b**) Percent oxidation resistance, after exposure to Fenton’s reagent.

**Figure 4 membranes-15-00282-f004:**
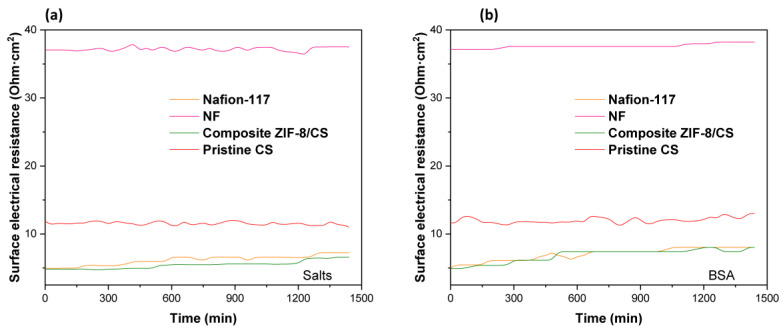
(**a**) Variation in the surface electrical resistance of membranes in the presence of salts (NaCl and Na_2_SO_4_). (**b**) Change in the surface electrical resistance of the membranes in the presence of the organic pollutant BSA.

**Figure 5 membranes-15-00282-f005:**
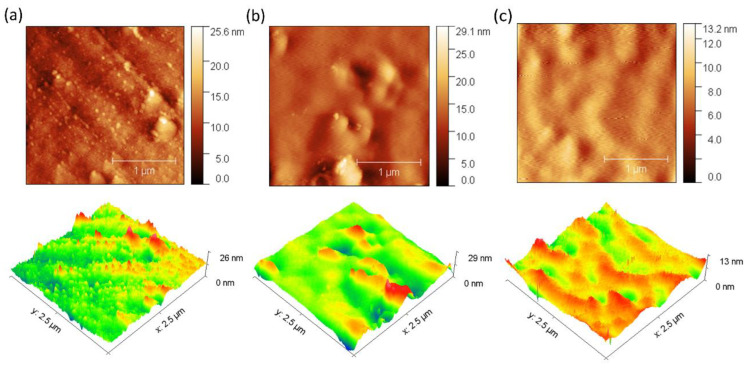
AFM images of the clean membranes: (**a**) Nafion-117. (**b**) NF. (**c**) Composite ZIF-8/CS.

**Figure 6 membranes-15-00282-f006:**
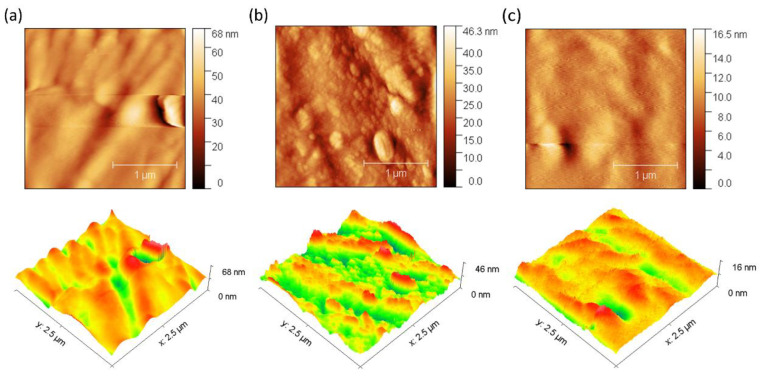
AFM images of the dirty membranes: (**a**) Nafion-117. (**b**) NF. (**c**) Composite ZIF-8/CS.

**Figure 7 membranes-15-00282-f007:**
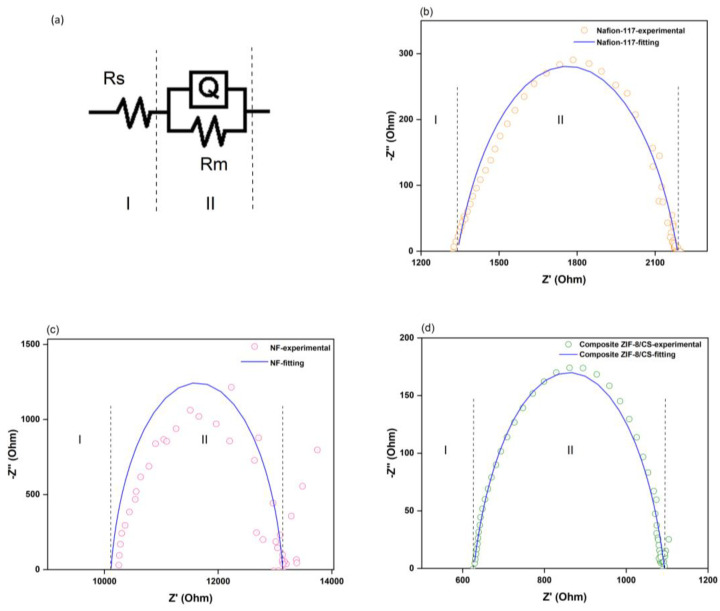
(**a**) Equivalent circuit R(QR). (**b**) Nyquist plot of Nafion-117 membrane, experimental data and fit. (**c**) Nyquist plot of nanofiltration membrane, experimental data and fit. (**d**) Nyquist plot of composite ZIF-8/CS membrane, experimental data and fit.

**Figure 8 membranes-15-00282-f008:**
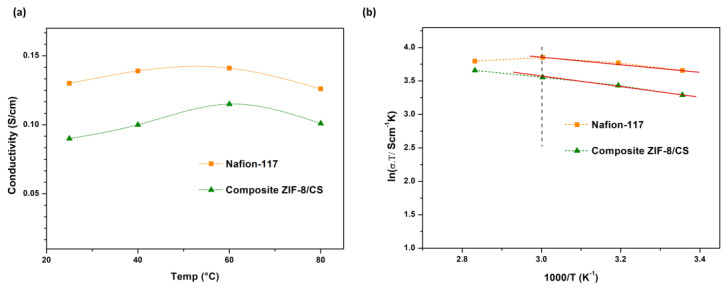
(**a**) Temperature dependence of the ionic conductivity of the membranes. (**b**) Arrhenius plot of the membranes: ln(σ·T) vs. 1000/T.

**Table 1 membranes-15-00282-t001:** Parameter estimates of mean roughness (Sa) and maximum height (Sz).

Parameter	Membranes		
	Nafion-117	NF	Composite ZIF-8/CS
Mean roughness (Sa, nm)	1.54 before → 4.30 after	1.98 before → 4.30 after	0.85 before → 1.07 after
Change in Sa (%)	179.48	117.12	25.88
Maximum height (Sz, nm)	22.10 before → 67.90 after	29.14 before → 46.35 after	13.05 before → 16.45 after
Change in Sz (%)	207.30	59.01	26.02

**Table 2 membranes-15-00282-t002:** Parameters estimated from the impedance spectroscopy results.

Parameter	Membranes		
	Nafion-117	NF	Composite ZIF-8/CS
Membrane resistance (Rm, Ω)	844.7	3023.0	465.8
Chi-squared (χ^2^, Fit)	1.7 × 10^−4^	6.9 × 10^−3^	8.2 × 10^−5^
conductivity (σ, Scm^−1^)	0.13	1.3 × 10^−2^	0.09

**Table 3 membranes-15-00282-t003:** Comparison of membranes for electrochemical systems. This study highlights the composite ZIF-8/CS membrane with high water uptake, low surface resistance, strong antifouling performance (against BSA and salts), and good ionic conductivity.

Study	Membrane	Thickness (µm)	Water Uptake (%)	Ionic Conductivity (S·cm^−1^)	Main Advantages
This work	PAN/Chitosan/ZIF-8	120	160.00	0.09	Low surface resistance, excellent antifouling properties (BSA and salts), and high oxidative stability
[58]	Nafion 117	200	15.87	0.10	Low surface resistance, good mechanical strength, and high oxidative stability
[59]	Chitosan/MWCNT	240	0.7	–	Biocompatibility, low surface resistance, mechanical and thermal stability
[60]	PVA/STA/GO	112	–	0.035	Mechanical strength, thermal stability, and hydrophilicity
[61]	Zr-MOF/PVDF	–	–	–	Improved ion selectivity and proton conductivity from Zr-MOF; PVDF provides chemical and mechanical stability
[62]	SPEEK/STA	190	21.28	1.54 × 10^−3^	High proton conductivity, oxidative stability, and hydrophilicity
[37]	Sulfonated PSEBS/ sulfonated SiO_2_	120	40	3.21 × 10^−2^	Excellent proton conductivity and membrane stability.

## Data Availability

The data supporting the findings of this study are available from the corresponding author upon reasonable request.

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
