# Peer review of "ZIF-8/Chitosan Composite Hydrogel as a High-Performance Separator for Bioelectrochemical Systems"

_membranes, 2025, doi:10.3390/membranes15090282_

Round 1
Reviewer 1 Report (New Reviewer)
Comments and Suggestions for Authors
This manuscript introduces a ZIF-8/chitosan composite membrane as a separator for bioelectrochemical systems (BES), demonstrating promising properties in terms of ionic conductivity, anti-fouling performance, oxidative durability, and mechanical integrity. The idea of reinforcing a natural biopolymer with a porous metal-organic framework is sound and aligns well with current trends in sustainable membrane materials. Moreover, benchmarking the composite membrane against Nafion-117 and nanofiltration membranes is highly relevant.
However, the current manuscript suffers from several issues that need to be addressed before it can be considered for publication. These include insufficient mechanistic interpretation, lack of statistical rigor, and overly descriptive data analysis in some sections. Specific concerns are outlined below.
- Line 50–68 The discussion of the limitations of Nafion-117 and NF membranes is useful, but highly generalized. Please include specific data (e.g., degradation threshold pH or cost comparison per m²) to support your claims.
- Line 74–81 The novelty of the ZIF-8/CS membrane should be better articulated. Has ZIF-8 previously been applied in BES separators? If so, what distinguishes your approach from existing studies?
- Line 110–122 The membrane fabrication lacks reproducibility details: what is the casting volume or thickness of the wet film? Was the PAN support pre-treated in any way?
- Line 144–156 Please include scale bars and magnification in all SEM and AFM figures, and clarify whether cross-section imaging was conducted on fractured or microtomed samples.
- Line 202–210 Water absorption is presented without standard deviation or replicate information. Please provide n values and error bars for statistical credibility.
- Line 222–230 The oxidative stability comparison is convincing. However, the 96% stability of the ZIF-8/CS membrane still implies 4% weight loss. Did SEM or FTIR show any degradation post-exposure?
- Line 238–248 Please specify how surface resistance was normalized—was it per unit membrane area or thickness? A control test without membrane is mentioned but not shown.
- Line 325–335 The discussion could be strengthened by providing activation energy for proton transport or temperature-dependent conductivity tests. This would help validate the proposed transport mechanism.
Author Response
Dear Reviewer:
All authors appreciate the detailed review and valuable comments made on our manuscript entitled:
" ZIF-8/chitosan Composite Hydrogel as a High-performance Separator for Bioelectrochemical systems"
We carefully revised the manuscript and responded point by point to each remark.

Reviewer 2 Report (New Reviewer)
Comments and Suggestions for Authors
I appreciate the effort the authors have put into this work to make ZIF-8/chitosan composite membranes. However, the novelty of the study should be more clearly emphasized. The manuscript would benefit from a clearer articulation of what sets this work apart from existing literature in the field.
- Please clarify the surface area of the ZIF-8 used, as it is a critical parameter for understanding its contribution to membrane performance.
- The mechanism by which the composite enhances conductivity should be explained in more detail. It would be helpful to understand whether the improvement is due to increased ionic pathways, better water retention, or any specific interaction at the interface.
- It is unclear whether the conductivity measurements were conducted in-plane or through-plane. This distinction is important for accurately evaluating the results and comparing them with other studies.
- The explanation of the AFM images could be expanded. Specifically, it would be helpful to clarify how the observed black and yellow regions contribute to the membrane’s function or performance.
- Please include the active area of the membrane used in performance testing
- A comparison table summarizing this work alongside previous studies would be valuable in highlighting the improvements achieved and supporting the novelty of the approach.
- As a suggestion, plotting the equivalent circuit (R(QR)) Nyquist plots of the different membranes on a single graph could improve clarity and facilitate comparison between samples.
Author Response
Dear Reviewer:
All authors appreciate the detailed review and valuable comments made on our manuscript entitled:
" ZIF-8/chitosan Composite Hydrogel as a High-performance Separator for Bioelectrochemical systems"
We carefully revised the manuscript and responded point by point to each remark.

Reviewer 3 Report (New Reviewer)
Comments and Suggestions for Authors
The study reads well and its appropriate for the publication in the journal.
Author Response
We would like to sincerely thank the reviewer for their positive evaluation of our manuscript. We are pleased to hear that the study is considered appropriate for publication in Membranes. We have no further modifications to make and are grateful for the constructive review process.
Round 2
Reviewer 1 Report (New Reviewer)
Comments and Suggestions for Authors
The authors have addressed the comments raised by the reviewers very well.
This manuscript is a resubmission of an earlier submission. The following is a list of the peer review reports and author responses from that submission.
Round 1
Reviewer 1 Report
Comments and Suggestions for Authors
In this work, ZIF-8/chitosan (CS) composite membranes were developed for use as separators in bioelectrochemical systems. However, the lack of a thorough explanation of the methodology and misinterpretation of the material properties make it difficult for readers to understand the results. Additionally, the experimental data presented contradict the claims made in the manuscript. The following issues require careful consideration:
- The membrane preparation process described in Section 2.3 of the manuscript is confusing. Does the composite ZIF-8/CS membrane include the PAN membrane? If not, why do the authors provide the XRD pattern and FTIR spectrum of PAN in Figure 2b and 2d? Moreover, as shown in Figure 2b, the XRD pattern of the composite membrane is identical to that of PAN, suggesting that PAN is the main component of the composite ZIF-8/CS membrane. If this is the case, naming the composite membrane as a “ZIF-8/CS composite hydrogel” would be inappropriate. However, the FTIR spectrum of the composite membrane in Figure 2d does not exhibit the characteristic peaks corresponding to PAN’s functional groups, indicating that PAN is not the main component of the composite membrane. The results from Figure 2b and 2d are contradictory.
- The samples lack basic specific parameters, such as the thickness of composite membrane.
- It is difficult to prove the wettability of the material to liquids only through water absorption experiments, and should be supplemented with experiments such as solid-liquid interfacial contact angle, porosity and other tests.
- Figure 2 (c) lacks a scale bar. More importantly, it is diffcult to observe the well-dispersed state of ZIF-8 within the CS matrix in this image, as noted by the authors in Section 3.1.2 of the manuscript.
- The analysis regarding the excellent compatibility and homogeneous interfacial dispersion between the ZIF-8 nanofiller and the CS matrix lacks sufficient description. Such a description is essential to elucidate the high stability of the sample.
- The experiments investigating the antioxidant properties of the materials lacked sufficient evidence. Additional validation experiments and relevant supporting data should be provided to substantiate the findings.
- Figures 5 and 6 show the changes of the membranes' surface morphology before and after exposure to clean and polluted conditions, demonstrating its anti-pollution properties. However, the specific anti-pollution mechanism is not explained.
Author Response
Dear Reviewer:
I appreciate the detailed review and valuable comments made by you on our manuscript entitled:
" ZIF-8/chitosan Composite Hydrogel as a High-performance Separator for Bioelectrochemical systems"
We have carefully revised the article and responded point by point to each remark. These improvements have strengthened the content and clarity of the paper. The corrections have been marked with red color in the revised manuscript.
We hope that the new manuscript meets the criteria for publication in Membranes.
I look forward to your comments and thank you for your time and consideration

Reviewer 2 Report
Comments and Suggestions for Authors
This manuscript reported a ZIF-8/chitosan (CS) composite membrane as a potential alternative to commercial membranes. The composite membrane exhibited comparable performance to Nafion 117 including ionic conductivity, surface resistance and anti-fouling properties. However, we believe that this manuscript does not demonstrate significant scientific advances and lacks a mechanistic understanding of the role of micro-porous ZIF-8 particles in the composite membrane. Therefore, we do not recommend this manuscript for th publication in Membranes in its current form. Below are main concerns:
- The properties of the composite membrane should be vary depending on the ratio of ZIF-8 to CS, so the optimization results is highly recommended. Further, control experiment with CS membrane (w/o ZIF-8) is missing. These are essential for understanding the role of ZIF-8 to the membrane performance.
- The characterization data presented in Figure 2 (b-d) are insufficient to confirm the presence and successful incorporation of ZIF-8 in the composite membrane. What is the amount of ZIF-8 above the composite membrane?
- ZIF-8 is generally know to be hydrophobic due to the methyl functional group in the Hmim linker. What is the reason on the high water uptake on the composite membrane? Water contact angle measurement could be useful to analyze the hydrophillicity of the composite membrane.
- Based on the surface resistance measurement (Figure 4) and AFM analysis (Figure 5-6), there are no significant degradation among the membranes. Since no significant degradation are observed with BSA across all membranes, it is hardly to conclude that composite membrane possess effective anti-fouling property.
Additional comments:
- In P2 L50, the statement that "because they can become protonated in alkaline environments, ~ " is incorrect. Sulfonic acid group are typically deprotonated under above mentioned condition.
- Undefined abbreviations:
- P1 L37: MFC
- P2 L77: CS
- P3 L111: PAN
- Typos and formatting errors:
- P3 L116: etanol
- Caption of Figure 1: (AEM) Anion-exchange membrane (+y-)
- P8 L222: FITR
Author Response

(The authors gave the same response as above.)

Reviewer 3 Report
Comments and Suggestions for Authors
This study explores the development of chitosan biopolymer membranes reinforced with channel-selective ZIF-8 nanofillers for Bioelectrochemical Systems (BES). The research design is well-constructed, the data is comprehensive, and the conclusions are generally reliable. However, there are several areas that require further clarification and refinement:
- The unique mechanism underlying the synergistic effect between ZIF-8 and chitosan needs further elaboration. For example, how the micropores of ZIF-8 contribute to enhancing the ion selectivity of chitosan should be clearly explained. Additionally, recent advances in related research and the key challenges in this area should be discussed.
- While the ion conductivity of the composite membrane after BSA treatment is comparable to that of Nafion-117, the long-term stability of this performance has not been addressed. A discussion on the long-term stability and potential performance degradation over time would be valuable.
- The study mentions that the composite membrane exhibits superior antifouling performance compared to Nafion-117, but the reasons behind this improvement need to be analyzed in greater depth, particularly in terms of the membrane's surface chemical properties.
- The parameters estimated from the EIS analysis, as presented in Table 2, appear to be inconsistent with the EIS spectra. It is recommended to provide a more detailed explanation of the calculation process.
- To further validate the practical application potential of the composite membrane, it is suggested to include performance testing in real BES environments, such as evaluating the power generation efficiency of microbial fuel cells.
Author Response

(The authors gave the same response as above.)

Reviewer 4 Report
Comments and Suggestions for Authors
The article "ZIF-8/chitosan Composite Hydrogel as a High-performance Separator for Bioelectric systems" by Henry Pupiales, Raúl Bahamonde Soria, Arboleda Daniel, Cevallos Carlos, Christian Alcivar, Laurent A. Francis, Xiao Xu, Patricia Luis is devoted to the study of a novel separator based on a chitosan biopolymer in combination with nanochannels created from ZIF-8 nanofillers. This is a very interesting direction in creating environmentally attractive membranes for various applications. The authors used the necessary techniques to characterize the obtained membranes and to compare them with the commercial membrane Nafion 117.
However, I have the following questions and comments about this work.
- Why do the authors perform the comparisons prepared membranes namely with Nafion 117? It seems that it should be the reinforced membrane, for example, Nafion XL.
- The geometricdimensions of the obtainedmembranesshould be indicatedin the article.
- Diffraction from a steel substrate prevails in the X-ray spectra of the ZIF-8/CS composite (Fig. 2b). It is difficult to understand anything from these data, and the authors' conclusion "...demonstrating that the incorporation of ZIF-8 nanofillers did not influence the CS crystal form" seems unpersuasive. It is necessary to carry out X-ray studies on another substrate which could avoid these problems.
- The authors claim that the impedance spectra were analyzed on the basis of an equivalent circuit (Fig.1C) using the ZSimpWin program. In this regard, it is necessary to give an example of a fitting spectrum and indicate the values of Rs, Zb and Zf on this Figure.
- The sense of Ag/AgCl electrodes applications for the impedance studies is not clear. The silver ions can be injected into the sample resulting in ambiguous interpretation of the results. It is necessary to carry out measurements using Pt electrodes, where such processes are absent.
- Authors wrote that: “The Nyquist plots show in Figure 7 have two distinctive parts, at high frequencies a resistance-capacitor loop, meanwhile at low frequencies a Warburg element is observed.”
However, no any Warburg element was included to the equivalent circuit for fitting. What does it mean?
The article should be essentially improved and the indicated studies (X-ray and impedance) should be redone.
Author Response

(The authors gave the same response as above.)

Round 2
Reviewer 2 Report
Comments and Suggestions for Authors
In this revised version, I think that the authors have carefully made the correction and response according to the Reviewers' concerns, and thus suggest accepting this work at its current state.
Author Response
Agradecemos sinceramente al revisor sus comentarios positivos y alentadores. Nos complace saber que las revisiones han abordado las inquietudes planteadas y agradecemos enormemente su recomendación de aceptación.
Reviewer 3 Report
Comments and Suggestions for Authors
The manuscript can be accepted in the present form.
Author Response
We sincerely thank the Reviewer for the positive and encouraging feedback. We are glad to know that the revisions have addressed the concerns raised, and we greatly appreciate your recommendation for acceptance.
Reviewer 4 Report
Comments and Suggestions for Authors
The authors have revised the article in many parts, but I have some serious questions and comments about this version. The example of the impedance spectrum fitting given in response to my comments should be included in the article so that the reader can see the correctness of the calculations. The values Rs, Zb and Zf should be indicated in Figure 7 (see, for example, https://iopscience.iop.org/article/10.1149/2.0571414jes ). I asked to do it in the previous review report.
New comments. The data shown in Table 1 contains errors in the calculation of the specific conductivity. For example, for Nafion 117, the specific conductivity is 0.018/(50.76 x 1.76) ~ 0.0002 S/cm, whereas it is indicated as 0.00056 S/cm in the Tab.1.
But it results in a more important question. This value of the specific conductivity (even indicated in the Tab.1) is very small for Nafion in the hydrated state. According to the specification and numerous studies, it should be order ~0.1 S/cm. Therefore, either the authors incorrectly determined the membrane resistance, or the membrane is not fully hydrated. In the last case it is necessary to specify the amount of water content (lambda) and explain why measurements are needed with such a lambda.
There is also a question about the quality of the added ZIF-8/CS X-ray spectrum. Why are only CS reflexes visible on the ZIF-8/CS composite film? Why these reflexes have a higher intensity than that of pure CS material?
A remark left over from my previous review. Authors cannot write about the Warburg element if it is not used in the description of the impedance spectrum. The CPE and Warburg impedance elements have really mathematically similar forms, but physically characterize different processes. In addition, no Warburg loop has been experimentally observed by the authors and this mention only confuses. Authors should clarify this part of the manuscript.
Moreover, it generally seems doubtful to correctly determine the membrane impedance response using only two short arcs from the impedance spectrum. The authors should present the evidences of proper impedance spectra interpretation. For example, by the measurements of the membrane with different geometric parameters (surface area or thickness) or measurements with different types of electrodes. It allows one to understand the frequency region of the impedance spectrum responsible for the bulk membrane response and the frequency range corresponding to the interface impedance.
Unfortunately, I again cannot recommend the article for the publication due to presence of serious errors and incorrections.
Author Response
Dear Reviewer:
All authors appreciate the detailed review and valuable comments made on our
manuscript entitled:
" ZIF-8/chitosan Composite Hydrogel as a High-performance Separator for
Bioelectrochemical systems"
We carefully revised the manuscript and responded point by point to each remark.

Round 3
Reviewer 4 Report
Comments and Suggestions for Authors
Dear authors!
You write in your reply: “Water content (λ) was not directly measured in this work…”
It is unclear how to use the conductivity measurement results presented in the article without monitoring of the sample state. It is also impossible to understand to what extent the membranes developed by the authors are interesting for application in various electrochemical cells.
To my opinion, authors use an incorrect method for the specific conductivity determination of membranes. The thickness of the studied membranes is l=0.018 cm for Nafion 117 (https://www.fishersci.se/shop/products/nafion-r-n-117-membrane-0-180mm-thick-0-90-meq-g-exchange-capacity-thermo-scientific/11389827 ) and the thickness of fabricated membranes is l~0.0043 cm. But the authors use the distance between the measuring electrodes d=0.5 cm much more than l. The medium between the sample and the electrode also affects the measured impedance spectrum.
The formula for the specific conductivity sigma= 1/R x (l/S) is valid for a homogeneous electric field, where l is the thickness of the sample, and S is the area of the electrodes directly applied to the sample. Thus, the geometric factor d/S is completely non correct for your case.
Once again, I insist on measuring samples of the same composition with different thicknesses in order to experimentally observe the direct proportionality of the membrane Zb resistance (determined from the impedance spectra) versus the thickness l of the sample.
The authors write:
"The frequency shift observed in this low-frequency region suggests that Nafion-117 and the composite ZIF-8/CS membrane exhibit higher ionic conductivity and lower resistance (Zb) compared to the NF membranes..."
It contradicts the data given in the Table 1, which indicates the specific conductivity of the NF membrane is an order of magnitude higher than the conductivity of Nafion 117.
It is necessary to seriously improve the article. I cannot agree with the publication of the article in the present form. The authors should present reliable evidences of proper impedance spectra interpretation and correct calculations of the specific conductivity. It is not clear how the reader can use the presented data without hydration level control.
In principle, the authors develop the interesting direction in the membrane researches but to my regret this article version should be finally rejected.
Author Response

(The authors gave the same response as above.)
